

# The driving factors of new particle formation and growth in the polluted boundary layer

Mao Xiao[1], Christopher R. Hoyle[1,2], Lubna Dada[3], Dominik Stolzenburg[4], Andreas Kürten[5], Mingyi Wang[6], Houssni Lamkaddam[1], Olga Garmash[3], Bernhard Mentler[7]; Ugo Molteni[1], Andrea Baccarini[1], Mario Simon[5], Xu-Cheng He[3], Katrianne Lehtipalo[3,8], Lauri R. Ahonen[3], Rima Baalbaki[3], Paulus S. Bauer[4], Lisa Beck[3], David Bell[1], Federico Bianchi[3], Sophia Brilke[4], Dexian Chen[6], Randall Chiu[9], António Dias[10], Jonathan Duplissy[3,11], Henning Finkenzeller[9], Hamish Gordon[12], Victoria Hofbauer[6], Changhyuk Kim[13,14], Theodore K. Koenig[9], Janne Lampilahti[3], Chuan Ping Lee[1], Zijun Li[15], Huajun Mai[13], Vladimir Makhmutov[16], Hanna E. Manninen[17], Ruby Marten[1], Serge Mathot[17], Roy L. Mauldin[18,19], Wei Nie[20], Antti Onnela[17], Eva Partoll[7], Tuukka Petäjä[3], Joschka Pfeifer[5,17], Veronika Pospisilova[1], Lauriane L. J. Quéléver[3], Matti Rissanen[3‡], Siegfried Schobesberger[15], Simone Schuchmann[17], Yuri Stozhkov[16], Christian Tauber[4], Yee Jun Tham[3], António Tomé[21], Miguel Vazquez-Pufleau[4], Andrea C. Wagner[5,9‡‡], Robert Wagner[3], Yonghong Wang[3], Lena Weitz[5], Daniela Wimmer[3,4], Yusheng Wu[3], Chao Yan[3], Penglin Ye[6,22], Qing Ye[6], Qiaozhi Zha[3], Xueqin Zhou[5], Antonio Amorim[10], Ken Carslaw[12], Joachim Curtius[5], Armin Hansel[7], Rainer Volkamer[9,19], Paul M. Winkler[4], Richard C. Flagan[13], Markku Kulmala[3,11,20,23], Douglas R. Worsnop[3,22], Jasper Kirkby[5,17], Neil M. Donahue[6], Urs Baltensperger[1*], Imad El Haddad[1*] & Josef Dommen[1]

[1]Laboratory of Atmospheric Chemistry, Paul Scherrer Institute, 5232 Villigen, Switzerland
[2]Institute for Atmospheric and Climate Science, ETH Zurich, 8092 Zurich, Switzerland
[3]Institute for Atmospheric and Earth System Research (INAR) / Physics, University of Helsinki, 00014 Helsinki, Finland
[4]Faculty of Physics, University of Vienna, 1090 Vienna, Austria
[5]Institute for Atmospheric and Environmental Sciences, Goethe University Frankfurt, 60438 Frankfurt am Main, Germany
[6]Center for Atmospheric Particle Studies, Carnegie Mellon University, Pittsburgh, PA 15213, USA
[7]Institute of Ion Physics and Applied Physics, University of Innsbruck, 6020 Innsbruck, Austria
[8]Finnish Meteorological Institute, 00560 Helsinki, Finland
[9]Department of Chemistry & CIRES, University of Colorado Boulder, Boulder, CO 80305, USA
[10]CENTRA and FCUL, University of Lisbon, 1749-016 Lisbon, Portugal
[11]Helsinki Institute of Physics, University of Helsinki, 00014 Helsinki, Finland
[12]University of Leeds, LS2 9JT Leeds, United Kingdom
[13]Division of Chemistry and Chemical Engineering, California Institute of Technology, Pasadena, CA 91125, USA
[14]School of Civil and Environmental Engineering, Pusan National University, 46241 Busan, Republic of Korea
[15]Department of Applied Physics, University of Eastern Finland, 70211 Kuopio, Finland
[16]P.N. Lebedev Physical Institute of the Russian Academy of Sciences, 119991 Moscow, Russian Federation
[17]CERN, CH-1211, Geneva, Switzerland
[18]Department of Chemistry, Carnegie Mellon University, Pittsburgh, PA 15213, USA
[19]Department of Oceanic and Atmospheric Sciences, University of Colorado Boulder, Boulder, CO 80309, USA
[20]Joint International Research Laboratory of Atmospheric and Earth System Sciences, School of Atmospheric Sciences, Nanjing University, Nanjing, Jiangsu Province, China
[21]IDL-Universidade da Beira Interior, Covilhã, Portugal
[22]Aerodyne Research Inc., Billerica, MA 01821-3976, USA
[23]Aerosol and Haze Laboratory, Beijing Advanced Innovation Center for Soft Matter Science and Engineering, Beijing University of Chemical Technology, Beijing, China.
‡now at: Aerosol Physics Laboratory, Physics Unit, Faculty of Engineering and Natural Sciences, Tampere University, Tampere, Finland
‡‡now at: Department of Chemistry & CIRES, University of Colorado Boulder, Boulder, CO 80305, USA





*Correspondence to*: Urs Baltensperger (urs.baltensperger@psi.ch); Imad El Haddad (imad.el-haddad@psi.ch)

**Abstract.**

New-particle formation (NPF) is a significant source of atmospheric particles, affecting climate and air quality. Understanding the mechanisms involved in urban aerosols is important to develop effective mitigation strategies. However, NPF rates reported in the polluted boundary layer span more than four orders of magnitude and the reasons behind this variability subject of intense scientific debate. Multiple atmospheric vapours have been postulated to participate in NPF, including sulfuric acid, ammonia, amines and organics, but their relative roles remain unclear. We investigated NPF in the CLOUD chamber using

mixtures of anthropogenic vapours that simulate polluted boundary layer conditions. We demonstrate that NPF in polluted environments are largely driven by the formation of sulfuric acid-base clusters, stabilized by the presence of amines, high ammonia concentrations and lower temperatures. Aromatic oxidation products, despite their extremely low volatility, play a minor role in NPF in the chosen urban environment but can be important for particle growth and hence for the survival of newly formed particles. Our measurements quantitatively account for NPF in highly diverse urban environments and explain

its large observed variability. Such quantitative information obtained under controlled laboratory conditions will help the interpretation of future ambient observations of NPF rates in polluted atmospheres.

# 1 Introduction

New-particle formation (NPF) is an important atmospheric phenomenon, affecting both climate (Dunne et al., 2016) and air

quality (Guo et al., 2014). Extremely high NPF rates are frequently observed in the polluted boundary layer, although current understanding suggests that newly formed particles should be rapidly scavenged by the high concentration of preexisting aerosols (Kulmala et al., 2017). Different vapours have been postulated to participate in NPF, including sulfuric acid, ammonia (Kirkby et al., 2011; Kürten et al., 2016), amines (Almeida et al., 2013), and organics (Kirkby et al., 2016; Lehtipalo et al., 2018; Riccobono et al., 2014). The high NPF rates, believed to drive haze events in China (Guo et al., 2014), have been

associated with the nucleation of sulfuric acid ($H_2SO_4$) in the presence of amines (Yao et al., 2018). In contrast, at other urban locations (Kuang et al., 2008), reported NPF rates are several orders of magnitude lower at similar $H_2SO_4$ concentrations, despite high levels of condensable species able to grow newly formed particles. These conflicting observations are subject of intense scientific debate of late (Brean et al., 2020; Cai et al., 2020; Guo et al., 2020) and highlight the need to better understand the role of the different vapours and environmental parameters and quantify their relative contribution in new-particle

formation and growth in different polluted locations.



Here we determine the parameters controlling particle formation and growth under polluted boundary layer conditions in the CERN CLOUD chamber (Cosmics Leaving OUtdoor Droplets (Kirkby et al., 2011)). We investigated a complex mixture of $H_2SO_4$, ammonia, dimethylamine (DMA), $NO_x$, ozone, water and several anthropogenic volatile organic compounds (AVOCs: naphthalene (NAPH), 1,2,4-trimethylbenzene (TMB) and toluene (TOL)), at different temperatures. Organic vapours from the oxidation of anthropogenic precursors are expected to contribute to formation and growth but their role is not yet quantified.

## 2 Results

### 2.1 Nucleation rates

NPF rates in the atmosphere often exhibit a clear correlation with $H_2SO_4$ (Kuang et al., 2008; Paasonen et al., 2010; Yao et al., 2018), but chamber experiments show that $H_2SO_4$ and water alone cannot explain boundary layer NPF events (Kirkby et al., 2011). Figure 1 presents our measured particle formation rates (at 1.7 nm diameter, referred to as $J_{1.7}$) as a function of $H_2SO_4$, in the presence of various concentrations of ammonia, DMA, oxidised anthropogenic organic vapours, and $NO_x$, on top of ambient urban observations (sub-2 nm $J$). Though they are highly correlated with the $H_2SO_4$ concentration for otherwise fixed conditions, the CLOUD formation rates span more than five orders of magnitude for the same $H_2SO_4$ concentration at different conditions, similar to the ambient observations, both in their magnitude and dependence on $H_2SO_4$ (Kuang et al., 2008; Paasonen et al., 2010; Yao et al., 2018).

Most of the variation in the CLOUD measurements is driven by DMA. Despite much lower ambient concentrations compared to ammonia, amines can be a key driver of boundary layer NPF (Almeida et al., 2013; Yao et al., 2018). At 293 K, the addition of 4 pptv DMA to an $NH_3$ / $H_2SO_4$ mixture increases the particle formation rates by two to three orders of magnitude (magenta squares compared to open red triangles). At these relatively high temperatures, NPF rates do not reach the kinetic limit of $H_2SO_4$ nucleation (solid cyan line, Fig. 1), suggesting that higher amine levels could increase NPF rates further by 1 - 2 orders of magnitude.

In the absence of amines, $J_{1.7}$ is extremely sensitive to ammonia concentrations (Fig. 2A). Even with 1 - 2 ppbv $NH_3$ at 293 K (Fig. 1), atmospherically relevant $H_2SO_4$ concentrations result in NPF rates only near the lower end of those observed in the polluted boundary layer. These rates can increase by a factor of 100 when $NH_3$ rises to 10 ppbv (Fig. 2A), a level that is frequently found in polluted cities (Elser et al., 2018; Guo et al., 2017). Our new measurements at higher ammonia levels extend the previous parameterisation of Dunne et al. (2016). and reveal a $J_{1.7} \propto [NH_3]^2$ dependence within the atmospherically relevant range of $[NH_3]$, in agreement with kinetic nucleation modelling (see Materials and Methods) based on thermodynamic data (Kürten, 2019) derived from CLOUD measurements and quantum chemical calculations.

NPF rates strongly increase with decreasing temperature (Fig. 1). The formation rate of $H_2SO_4$-$NH_3$ particles increases by two orders of magnitude at 278 K (blue line) compared to 293 K (red line). A similar increase in the formation rates of $H_2SO_4$-DMA particles is observed with the same temperature decrease (cyan line at 278 K vs. magenta line at 293 K). Accordingly, NPF at 278 K proceeds close to kinetic limit with only 4 pptv of DMA, more than 4 orders of magnitude faster than our





baseline experiments (H$_2$SO$_4$ with 1 ppbv NH$_3$ at 293K, red dashed line). This agrees with $J_{1.7}$ values extrapolated by Kürten et al. (2018) from measurements at 3.2 nm in Almeida et al. (2013). We modelled the temperature dependence of $J_{1.7}$ for both

H$_2$SO$_4$-NH$_3$ and H$_2$SO$_4$-DMA systems (Fig. S1) and find that H$_2$SO$_4$-base NPF rates can be reasonably well simulated at both temperatures, based on experimental and quantum-chemically-calculated thermodynamic data (Kürten, 2019; Myllys et al., 2019).

To investigate the effect of organic vapours, we initiated the photo-oxidation of a mixture of AVOCs (naphthalene, 1,2,4-trimethylbenzene and toluene), in the presence of SO$_2$, NH$_3$ and 0.1 - 1.5 ppbv NO. $J_{1.7}$ increased by a factor of 2 - 100

compared to experiments without AVOCs (red filled triangles in Fig. 1A). Enhancements in NPF rates are significant in the absence of DMA, for the H$_2$SO$_4$-NH$_3$ system at 293 K (red dashed line), where NPF rates are lowest. Under these conditions, variations in the enhancement factors are explained by the NO$_x$ levels and the amounts of oxidised organics (OxOrg) (Fig. 2B-C, Fig. S2, Fig. S3). However, $J_{1.7}$ strongly decreases with increasing NO, since NO suppresses both autoxidation and the formation of low-volatility dimers. Compared with our baseline experiments, organics enhance $J_{1.7}$ by a factor of 20 at 0.2

ppbv NO, but only by a factor of 2 at 1.2 ppbv NO, i.e., $J_{1.7}$ is almost the same as for H$_2$SO$_4$ nucleating with NH$_3$ alone (Fig. 2D). It should also be noted that the HOMs dimer formation could be higher in chamber simulations like CLOUD than in the ambient atmosphere where extra HO$_2$ sources such as photolysis of aldehydes or OH reacting with CO or small oxygenated hydrocarbons are present (see SI for more details). In the presence of amines, the contribution of organics to NPF is marginal, since the inorganic nucleation rate is overwhelming. This is even more pronounced at higher NO mixing ratios, typical of

urban atmospheres, where the influence of organics is further diminished.

Fig. 3 recapitulates the changes in NPF rates ($J_{1.7}$) resulting from different factors (besides NH$_3$ and NO concentrations discussed in Fig. 2A and 2D) under fixed H$_2$SO$_4$ concentration. Lower temperature stabilizes nucleating clusters and enhances NPF rates under all conditions. While the addition of both DMA and organics leads to an enhancement in $J_{1.7}$, the effect of only 4 pptv DMA is much more significant. Further addition of organics when DMA is already present will only marginally

affect nucleation rates.

## 2.2 Growth of newly formed particles

In view of their high diffusivity, the growth of newly-formed particles up to 3 nm and above is crucial for their survival against scavenging loss. Condensation of H$_2$SO$_4$ and oxidised organics, as well as cluster coagulation contribute to growth (Lehtipalo et al., 2018). H$_2$SO$_4$ grows sub-3 nm particle by $(1.94\pm0.13)\times 10^{-7}$ nm h$^{-1}$ cm$^3$, insufficient for GR measured at urban location.

Such GR are consistent with the parameterisation in Lehtipalo et al. (2018) and model predictions including van-der-Waals interactions (Stolzenburg et al., 2020), while higher than estimates that assume the collision of hard spheres at the kinetic limit (Nieminen et al., 2010). DMA, low temperatures, and high NH$_3$ concentration (Lehtipalo et al., 2016) enhance molecular cluster concentration. When nucleation approaches the kinetic limit, cluster coagulation dominates the growth. As stable H$_2$SO$_4$–DMA clusters, formed to a great extent at 278 K, are lost less rapidly to pre-existing aerosols than are individual

molecules (monomers), these clusters increase the condensable H$_2$SO$_4$ species reservoir and, thereby, the H$_2$SO$_4$ contribution





to growth relative to organic vapours as seen in Fig. 4B (cyan squares). This increases the effective concentration of $H_2SO_4$ beyond the measured bare monomers, especially in the presence of high condensation sinks found in polluted environments. While NPF is dominated by the formation of $H_2SO_4$ clusters, anthropogenic organic vapours strongly contribute to growth, and are required to explain the GR measured at urban locations. The multicomponent system used in the simulations does

reproduce the range of GR found in urban atmospheres (4-12 nm $h^{-1}$ for $10^7$ $cm^{-3}$ $H_2SO_4$). The measured GR are consistent with a 0.5 - 3% yield of OxOrg that are able to condense onto particles in the sub-3nm size range (Fig. S4).

Intense NPF events have been observed under highly polluted conditions, despite the high loss rates of nucleated clusters to pre-existing particles. Here, we investigated the survival probability at condensation sinks up to 0.02 $s^{-1}$. We define the survival probability as the ratio $J_{2.5}/J_{1.7}$ (see Fig. S5 for $J_6/J_{1.7}$) and display $J_{2.5}/J_{1.7}$ against the condensation sink, $CS_6$, in Fig. 5, where

$CS_6$ only includes particles larger than 6 nm that are typically reported from scanning mobility particle sizer measurements. For the case of $H_2SO_4$-$NH_3$ nucleation, the survival probability decreases with increasing $CS_6$, and increases with increasing GR. When NPF rates approach the kinetic limit in the presence of DMA, as, for example, in Shanghai (Yao et al., 2018), collisions between the abundant nucleating clusters reduce the survival probability of sub-3 nm particles by an order of magnitude, but simultaneously increase the growth rate of those that do survive. The survival probability only becomes

dependent on the $CS_6$ when $CS_6$ is greater than the loss rate due to cluster-cluster collisions. For $10^7$ $cm^{-3}$ $H_2SO_4$, the measured survival probability of newly formed particles remains virtually independent of $CS_6$ even for values as high as 0.01-0.02 $s^{-1}$ (Figure 5, Fig. S5), consistent with model simulations (black line). This is much higher than previously estimated (Kulmala et al., 2017; Yao et al., 2018) when van-der-Waals interactions were not considered (Fig. S6), and explain the low dependence of particle formation on CS observed even in highly polluted urban environments. Above 0.02 $s^{-1}$, $CS_6$ becomes the dominant

sink of clusters, sharply decreasing their survival probability.

Figure 6 shows a mass defect plot of the nucleating species during an experiment with $H_2SO_4$, $NH_3$, DMA, $NO_x$, and oxidised anthropogenic organics at 278 K, representative of recent ambient measurements in Shanghai (Yao et al., 2018). Similar observations were also presented for the case of Beijing (Cai et al., 2020). In the same way as in these ambient observations, pure $H_2SO_4$ clusters are seen up to tetramers. Larger acid clusters contain the stronger base, DMA. While the precursor

composition in the ambient atmosphere is indeed much more complex than in the chamber, we demonstrate that the features of HOMs in both cases are very similar, representative of high $NO_x$ conditions. This includes the high abundance of HOMs with a molecular weight between 200-400 Th, no significant monomer/dimer patterns, and a high contribution of N-containing HOMs. These observations indicate that, under similar conditions, chamber simulations reproduce ambient observations.

## 2.3 *J* vs GRs

If the presence of amines (Cai et al., 2020; Yao et al., 2018) or organics (Brean et al., 2020; Guo et al., 2020) drives urban NPF has been an intense debate in recent atmospheric observations. While both organics and amines can enhance $H_2SO_4$-$NH_3$ nucleation to different extents (Fig. 3), they do so by different mechanisms. Amines stabilize the nucleating clusters while organics contribute to higher concentrations of condensing material. We compare chamber simulations and atmospheric



observations in Fig. 7 using sub-3 nm GR as a proxy for total condensable vapour concentrations, as most field observations
in urban atmospheres do not report the nucleating species at a molecular level. For $H_2SO_4$–base nucleation, $J_{1.7}$ and sub-3 nm
GR are tightly correlated through their dependence on $H_2SO_4$ concentration. At a fixed GR, $J$ values are enhanced by lower
temperatures and by DMA compared with ammonia. In line with this, measurements by Yao et al. (2018) in Shanghai fall on
the $H_2SO_4$ kinetic limit nucleation line. Shanghai NPF shows no seasonal variation, implying that DMA is present year round
and that concentrations during the warm seasons are higher than 4 pptv. NPF during wintertime in Nanjing (Yu et al., 2016)
also falls on the $H_2SO_4$ kinetic limit nucleation line, and so is likely driven by amines as well. The CLOUD measurements of
NPF at 278 K with $H_2SO_4$, 1-2 ppbv ammonia and organics compare well with atmospheric observations in several polluted
environments during cold seasons (Beijing winter; Jayaratne et al., 2017) and Yangtze River Delta region (Dai et al., 2017))
implying low amine concentrations (< 4 pptv) in these regions in winter. Atmospheric observations in warm environments lie
well above the $J$ vs GR line for 1 to 2 ppbv ammonia nucleating with $H_2SO_4$ at 293 K. Ammonia concentrations of ~10 ppbv
could explain the observations from Madrid (Carnerero et al., 2018), Po Valley (Kontkanen et al., 2017) and Tecamac (Kuang
et al., 2008), while unrealistically high ammonia concentrations (>100 ppbv) would be needed to explain Beijing spring (Cai
et al., 2017) or Nanjing summer (Yu et al., 2016). Therefore, NPF in the latter two cases is likely due to amines at pptv levels,
which would match the CLOUD measurements well (magenta squares in Fig. 7). Looking at the variability in NPF observed
in Shanghai, Nanjing (summer and winter) and the Yangtze River Delta, we see evidence for strong regional variation in
polluting vapours.

## 3 Conclusions

Urban NPF rates are primarily driven by the ambient concentrations of $H_2SO_4$ and the presence of stabilising base vapours.
The largest effect results from amines, if present, as even low pptv levels are sufficient for NPF rates to approach the kinetic
limit for $H_2SO_4$ particles. Reducing amine emissions may therefore be an efficient way to reduce the particle number
concentrations, although the particle mass concentrations will not be strongly affected since the available condensable vapours
will condense onto pre-existing particles. Although ammonia is less efficient for stabilizing $H_2SO_4$ clusters, reducing the $NH_3$
concentrations can also bring strong benefits in conditions where amine concentration is limited, due to the quadratic
dependence of NPF on $NH_3$. Lower temperatures also strongly stabilize clusters, suggesting that particle number
concentrations may be affected by a warming climate. In spite of the high VOC emissions in cities, the contribution of organics
to nucleation is likely to be limited by high co-emission of $NO_x$, although organic oxidation products remain important for
particle growth. In contrast, in suburban environments, an enhanced organic contribution to NPF is expected because of lower
$NO_x$ and base concentrations. We conclude that in highly polluted environments such as Asian megacities with high ammonia
and $NO_x$ concentrations and, especially, in the presence of amines, $H_2SO_4$–base nucleation is the prime driver of new particle
formation and is effective even in the presence of high condensation sinks near 0.02 s$^{-1}$.




## 4 Materials and Methods

The experiments are performed at CERN CLOUD chamber (Cosmics Leaving OUtdoor Droplets (Kirkby et al., 2011)) with a complex mixture of $H_2SO_4$, ammonia, dimethylamine (DMA), $NO_x$, ozone and several anthropogenic volatile organic compounds (AVOCs: naphthalene (NAPH), 1,2,4-trimethylbenzene (TMB) and toluene (TOL)), at 293K and 278 K, 60%. Organic vapours from the oxidation of anthropogenic precursors, besides their effect on human health (Daellenbach et al., 2020) are expected to contribute to formation and growth, similar to biogenic products (Lehtipalo et al., 2018), but their role is still largely unknown. The data were collected between October and November in 2016 and 2017

### 4.1 Chamber experiments

Experiments were performed at the CLOUD facility (Kirkby et al., 2011) at CERN during two intensive campaigns in 2016 (CLOUD11 campaign) and 2017 (CLOUD12 campaign). All experiments were carried out at atmospheric pressure with ultrapure air created from the evaporation of liquid nitrogen (Messer, 99.999%) and liquid oxygen (Messer, 99.999%), which were mixed in the gas volume ratio 79% and 21%, respectively. The $O_3$ was generated by exposing a small fraction of the air through a quartz tube surrounded by UVC lamps (wavelength <240 nm). Relative humidity was adjusted by passing the air through a Nafion® humidifier using ultrapure water (18 MΩ cm, Millipore Corporation). The required $SO_2$ mixing ratio was provided from a pressurized gas cylinder (CARBAGAS AG, 500 ppmv in $N_2$). The anthropogenic volatile organic compounds were added by flushing air through an evaporator containing naphthalene (Sigma Aldrich, 99%) or from gas bottles of 1,2,4-trimethylbenzene (Messer) and toluene (Messer). With all the instruments connected to the CLOUD facility, the total sampling flow rate was 270 L min$^{-1}$, resulting in a dilution lifetime of 1.6 h in the chamber. All experiments presented here were performed in the presence of ions from galactic cosmic rays (GCR) at normal atmospheric levels, i.e. without ion removal using field cage electrodes, except for data specified as neutral experiments, which includes data without ions where the field cage electrodes were set to ±30kV.

In the CLOUD11 campaign, experiments were performed at 293±1.5 K (referred to as 293 K) and 57±3 % RH. Approximately 1 to 2 ppbv $NH_3$ were injected into the chamber from an $NH_3$ gas bottle (100 ppm in $N_2$, CARBAGAS AG). A set of UV sources including a 4 W KrF excimer UV laser (UVX) at 248 nm and four 200 W Hamamatsu Hg-Xe lamps (UVH) at wavelengths between 250 and 450 nm with adjustable power were used to photolyse 40 to 100 ppbv $O_3$ to produce 1 x 10$^6$ to 1 x 10$^7$ OH cm$^{-3}$. Extra UVA at 385 nm was generated from a 400 W UVA LED saber (UVS3) to photolyse $NO_2$ to NO.

Three sets of experiments were performed during CLOUD11: (1) experiments to determine NPF rates from sulfuric acid and ammonia, (2) experiments to determine NPF rates from sulfuric acid and typical anthropogenic volatile organic compounds (AVOCs), i.e., naphthalene (NAPH), 1,2,4-trimethylbenzene (TMB) and toluene (TOL) with and without ammonia, and (3) experiments to determine NPF rates from sulfuric acid, ammonia and AVOC mixtures in the presence of $NO_x$. A typical





experiment started by turning on UV sources to trigger oxidation of AVOCs and SO₂. Formation rates ($J$) and growth rates (GRs) were determined by measuring the dynamics of the particle number size distribution (see below).

In the CLOUD12 campaign, experiments were performed at 278±1 K (referred to as 278 K) or 294± 1 K (referred to as 293 K) and 60 ± 5 % RH. 1 to 2 ppbv NH₃ were injected into the chamber from an NH₃ gas bottle (100 ppm in N₂, Messer). 4 pptv

dimethylamine (DMA, 1% in N₂, Messer) was injected in the last part of the campaign to study the effect of amines on NPF. In addition to O₃ photolysis, OH radicals were also produced by nitrous acid (HONO) photolysis in this campaign. The gaseous HONO was synthesised by continuous mixing of H₂SO₄ (Sigma Aldrich, 99%) with NaNO₂ (Sigma Aldrich, 99%) in a specially designed stainless-steel reactor. 2 to 20 ppbv HONO was injected into the CLOUD chamber and photolysed by UVS3 producing $1 \times 10^6$ to $1 \times 10^7$ OH together with O₃ photolysis by UVH and UVX as in CLOUD11.

Three sets of experiments were performed during CLOUD12 at 293K and 278K: (1) experiments in the presence of sulfuric acid, ammonia and AVOC mixtures; (2) experiments in the presence of sulfuric acid and DMA, and (3) experiments with sulfuric acid, DMA and AVOC mixtures.

It is expected that precursor VOCs must have a large carbon skeleton to be able to form highly oxygenated molecules (HOMs) contributing to NPF. Aromatics comprise roughly 20-40% of VOC emissions in urban environments (Baker et al., 2008;

Boynard et al., 2014; Li et al., 2019), while the largest fraction is from small alkanes and alkenes that are not relevant for our study. The AVOC mixtures used in this study represent the most common aromatic compounds in the urban boundary layer. Toluene is the most abundant aromatic compound. We have not included benzene, as it has lower emission rates than toluene, and is unlikely to contribute to NPF due its low reactivity. TMB was chosen as representative of more reactive C2 and C3 aromatics, which individually have a factor of 1-5 smaller concentrations than toluene. Naphthalene is representative of

polycyclic aromatic compounds, and, potentially, of other unspeciated organics (e.g., long-chain alkanes (Bruns et al., 2016; Jathar et al., 2014)) which are emitted at lower rates but have high potential of forming SOA (Chan et al., 2009; Li et al., 2016) and HOMs (Molteni et al., 2018). We tested different concentrations of the three aromatics with different mixing ratios to represent the wide range of HOM forming organics in urban environments listed in Table S1.

## 4.2 Particle Measurements

Several independent instruments measured particle concentrations and number size distributions in the chamber. Number concentrations of freshly formed particles at several different cut-off sizes between 1 and 3 nm were determined by a particle size magnifier (Vanhanen et al., 2011) (PSM, Airmodus Ltd.) using diethylene glycol as working fluid. The PSM was used in scanning mode and the particle counting was done by a condensation particle counter (CPC) downstream of the PSM. A butanol ultra-fine CPC (TSI 3776) with fixed cut-off size measured the total particle number concentration above ca. 2.5 nm.

The particle size distribution between 1.8 and 8 nm was measured with a DMA-train (Stolzenburg et al., 2017) which consists of six differential mobility analysers and CPCs in parallel. For the two channels measuring sub-2.5 nm particles, a PSM was installed upstream of the particle counter. A commercial nano-SMPS (TSI 3938) with a water-CPC (TSI 3788) was used to measure the size distribution between 4.6 and 60 nm, and a custom-built SMPS, consisting of a krypton source, a long DMA



and a CPC (TSI 3010) was used to cover the size distribution between 20 and 400 nm. A neutral cluster and air ion spectrometer

(Mirme and Mirme, 2013) (NAIS, Airel Ltd.) measured ions with mobility diameters between 0.75 and 40 nm. A corona charger was periodically applied to detect particle size distributions in the size range of 2-40 nm using the NAIS.

### 4.3 Determination of new particle formation rates

Particle formation rates ($J$) were calculated as in a previous study (Dada et al., 2020):

$$J = \frac{dN}{dt} + S_{dil} + S_{wall} + S_{coag} \tag{1}$$

where $N$ is the particle number concentration above a certain cut-off size ($d_p$), determined by the PSM for 1.7 nm, the CPC for

2.5 nm and the nano-SMPS for 4.6 nm. The cut-off size has an uncertainty of approximately 0.5 nm.

Dilution was corrected for by applying the relation $S_{dil} = k_{dil}N$, where $k_{dil} = 1.72 \times 10^{-4} \, s^{-1}$ accounts for a 270 L min$^{-1}$ continuous flow into the 26.1 m$^3$ chamber.

Particle diffusion loss to the chamber wall was calculated as:

$$S_{wall} = \sum_{d'_p=d_p}^{d_{p,max}} N_{d'_p} k_{wall}(d'_p, T) \tag{2}$$

where wall loss rates were extrapolated from the sulfuric acid wall loss rate:

$$k_{wall}(d'_p, T) = 2.116 \times 10^{-3} \left(\frac{T}{278 \, K}\right)^{0.875} \left(\frac{0.82 \, nm}{d'_p}\right) \tag{3}$$

Coagulation losses were calculated as:

$$S_{coag} = \sum_{d_{p,i}=d_p}^{d_{p,max}} \sum_{d_{p,j}=d_{p,i}}^{d_{p,max}} \delta_{ij} K_{ij} N_i N_j \tag{4}$$

where $K_{ij}$ is the coagulation coefficient for particles of size $d_{p,i}$ and $d_{p,j}$, and $N_i$ and $N_j$ are the particle concentrations in the size bins $i$ and $j$. $\delta=1$ for $i \neq j$ and $\delta=0.5$ for $i=j$. Combined size distributions based on nano-SMPS and SMPS were used to calculate coagulation losses. For some experiments in CLOUD12 nano-SMPS measurements were not available. Instead, NAIS measurements in particle mode (i.e., with corona charger), cross validated with nano-SMPS measurements, were used to

construct size distributions for particles below 20 nm.

The $J$ values reported here are the median values of each experiment after reaching stable conditions. The uncertainty of $J$ (as error bars in the figures) was calculated by an error propagation method, accounting for the statistical variation of $J$ in the experiment and run-to-run repeatability of $J$ in CLOUD, which is ca. 30%.

Growth rates were determined by the appearance time method (Lehtipalo et al., 2014, 2016) using the 50% appearance time

of the DMA-train measurements from 1.8 nm to 3.2 nm.



### 4.4 Gas phase measurements

Gas monitors were used to measure ozone ($O_3$, Thermo Envrionmental Instruments TEI 49C), sulfur dioxide ($SO_2$, Thermo Fisher Scientific Inc. 42i-TLE) and nitric oxide (NO, ECO Physics, CLD 780TR). Nitrogen dioxide ($NO_2$) was measured by a cavity attenuated phase shift nitrogen dioxide monitor (CAPS $NO_2$, Aerodyne Research Inc.) and a custom-made cavity enhanced differential optical absorption spectroscopy (CE-DOAS) instrument. Relative humidity of the chamber was determined by dew point mirrors (EdgeTech).

Two $HNO_3$-chemical ionisation mass spectrometers (Jokinen et al., 2012) ($HNO_3$-CIMS) were connected to the chamber for gas phase measurements. One instrument was an atmospheric pressure interface time of flight mass spectrometer (Junninen et al., 2010) (APi-TOF) equipped with a commercial inlet (Airmodus, Ltd) using $NO_3^-$ as primary ions. Another $HNO_3$-CIMS with an ion precipitator in front of the inlet used a corona discharge as ion source and 50 ns reaction time. Sulfuric acid was measured by these $HNO_3$-CIMS and quantified following the same calibration and loss correcting procedures described in previous publications (Kirkby et al., 2016; Tröstl et al., 2016). The sulfuric acid concentrations measured by the two instruments agreed with each other within 20%. To avoid interference from charged ions in the chamber, the sulfuric acid concentrations from the $HNO_3$-CIMS with an ion precipitator were used in the study.

Ammonia concentrations were measured with an $H_3O^+$ chemical ionisation mass spectrometer ($H_3O^+$ CIMS) and a PICARRO $NH_3$ analyser. The $H_3O^+$ CIMS was an APi-TOF equipped with a crossflow inlet using positively charged water clusters to detect ammonia in real time. Dimethylamine was also measured by the $H_3O^+$ CIMS applying the calibration factor determined from ammonia. The measured concentration of dimethylamine (DMA) was in good agreement with estimations based on DMA injection rates. DMA time traces could also be measured by the $HNO_3$-CIMS (Simon et al., 2016), which was used for experiments when the DMA signal in the $H_3O^+$ CIMS was influenced by fragments from organics.

A proton transfer time of flight mass spectrometer (PTR3) (Breitenlechner et al., 2017) with a core sampling inlet, which transfers sample air through a tripole reaction chamber operated at 80 mbar, was used to measure the TMB and NAPH concentration. A quadrupole proton transfer mass spectrometer (PTR-MS with quadrupole, IONICON) measured the toluene concentration in CLOUD11 and an STOF proton transfer mass spectrometer (PTRS) was used in CLOUD12 to measure toluene, TMB and naphthalene. For some experiments in CLOUD11, the quadrupole PTR-MS measurements were not available, where we determined the toluene concentration from the injection rates (MFC settings).

### 4.5 Determination of OH and OxOrg concentrations

The OH concentration was derived from the sulfuric acid concentration as well as from the amount of TMB reacted. To determine OH from the sulfuric acid concentration, first order sulfuric acid change rates were fitted with a 15-min time window. A sulfuric acid production rate was then obtained by subtraction of sulfuric acid losses to the chamber wall and to particles. The OH concentration was determined as:





$$[OH] = \frac{P_{SA}}{k_{SA}[SO_2]} \qquad (5)$$

where $k_{SA}$=8.6 x 10$^{-13}$ cm$^3$ s$^{-1}$.

The OH concentration was also determined by fitting the TMB time series using:

$$\frac{d[TMB]}{dt} = k_{inj} - k_{dil} + k_{TMB}[OH][TMB] \qquad (6)$$

where $k_{inj}$ is the injection rate of TMB and $k_{dil}$ is the chamber dilution rate. The reaction rate constant for OH and TMB ($k_{TMB}$) is 3.25 x 10$^{-11}$ cm$^3$ s$^{-1}$. OH concentrations calculated from the two methods agreed within 20% at 293 K without DMA addition. For other conditions, OH concentration estimates from the two methods correlated with each other but the method using sulfuric acid concentration underestimated the OH concentration as the formation of sulfuric acid clusters was not taken into consideration. Therefore, OH concentrations from the sulfuric acid method were corrected by applying correction factors based on the OH estimation from the TMB time series. In the analysis here, the OH time series from the sulfuric acid concentration were used.

Oxidised organics (OxOrg) concentrations were estimated with

$$\frac{d[OxOrg]}{dt} = k_{VOC}[OH][VOC] - (k_{dil} + k_{wall} + CS)[OxOrg] \qquad (7)$$

The reaction rate constants $k_{VOC}$ are 2.6 x 10$^{-11}$ cm$^3$ s$^{-1}$ for NAPH, 3.25e x 10$^{-11}$ cm$^3$ s$^{-1}$ for TMB and 5.8 x 10$^{-12}$ cm$^3$ s$^{-1}$ for toluene. Dilution, wall loss and condensation to particles are considered by $k_{dil}$, $k_{wall}$ and CS assuming oxidised organics will not evaporate after they condensed on a wall or particles, which is a reasonable assumption for compounds relevant for nucleation and early growth.

### 4.6 Nucleation Model

A kinetic model based on the general dynamic equation (Seinfeld and Pandis, 2016) was used in order to assess acid-base nucleation. The model considers neutral clusters containing up to 15 sulfuric acid molecules and 50 geometric size bins between clusters containing 15 sulfuric acid molecules and 300 nm and uses a constant sulfuric acid production rate:

$$\frac{dN_{A_1B_0}}{dt} = Production + \sum \gamma_{A,A_kB_j}N_{A_kB_j} + \gamma_{A,A_2B_0}N_{A_2B_0}$$

$$- N_{A_1B_0}\left(k_{dil} + k_{wall,A_iB_i} + \sum_{j=1}^{\infty} K_{A_iB_i,j}N_j\right) \qquad (8)$$

Evaporation was considered for clusters containing less than 5 sulfuric acid molecules. Clusters containing more bases than acids were ignored as they are highly unstable.


$$\frac{dN_{A_iB_i}}{dt} = \frac{1}{2}\sum_{\substack{A_j+A_k=A_i \\ B_j+B_k=B_i}} K_{j,k}N_{A_jB_j}\,N_{A_kB_k} + \gamma_{A,A_i+1B_i}N_{A_i+1,B_i} + \gamma_{B,A_iB_i+1}N_{A_iB_i+1}$$

$$- N_{A_iB_i}\left(k_{dil} + k_{wall,A_iB_i} + \sum_{j=1}^{\infty} K_{A_iB_i,j}N_j\right) - (\gamma_{A,A_iB_i} + \gamma_{B,A_iB_i})\,N_{A_iB_i} \tag{9}$$

Clusters containing 5 to 15 sulfuric acid molecules are considered as stable and were treated the same regardless the base number.

$$\frac{dN_i}{dt} = \frac{1}{2}\sum_{j=1}^{i-1} K_{j,i-j}N_j\,N_{i-j} \; - N_i\left(k_{dil} + k_{wall,i} + \sum_{j=1}^{\infty} K_{i,j}N_j\right) \tag{10}$$

Then 50 geometric sized bins were applied, and the particles formed by collision of two particles were distributed between the
345   two nearest bins as given in Eq. (11) in order to link continuous growth to discrete collisions.

$$\frac{dN_i}{dt} = \frac{1}{2}\sum_{j+k\to i} K_{j,k}N_j\,N_kP_{j,k} + \frac{1}{2}\sum_{j+k\to i-1} K_{j,k}N_j\,N_k(1-P_{j,k}) - N_i\left(k_{dil} + k_{wall,i} + \sum_{j=1}^{\infty} K_{i,j}N_j\right) \tag{11}$$

Collision rates were calculated as:

$$K_{i,j} = \left(\frac{3}{4\,\pi}\right)^{\frac{1}{6}}\left(\frac{6k_BT}{m_i} + \frac{6k_BT}{m_j}\right)^{\frac{1}{2}}\left(V_i^{\frac{1}{3}} + V_j^{\frac{1}{3}}\right)^2 E_{i,j} \tag{12}$$

$$P_{j,k} = \frac{V_{i+1} - V_j - V_k}{V_{i+1} - V_i} \tag{13}$$

Where $V_i < V_j + V_k < V_{i+1}$.

$E_{i,j}$ is the collision enhancement factor due to the van-der-Waals force, which can be linked to the Hamaker constant ($H_A = 6.4 \times 10^{-20}$ J), as described by Chan and Mozurkewich (Chan and Mozurkewich, 2001).

$$E_{i,j} = 1 + \frac{\sqrt{A'/3}}{1 + 0.0151\sqrt{A'}} - 0.186\ln(1 + A') - 0.0163\ln^3(1 + A') \tag{14}$$

$$A' = \frac{4H_A d_{p_i}d_{p_j}}{kT(d_{p_i} + d_{p_j})} \tag{15}$$

350   where $d_{pi}$ and $d_{pj}$ are the diameters of the colliding particles. A density of 1.7 g cm$^{-3}$ was used for $(NH_4)_2SO_4$ in the calculations.
Evaporation of clusters containing less than five sulfuric acid molecules was taken into consideration using updated Gibbs free energies reported by Kürten (2019) based on previously published CLOUD results for ammonia - sulfuric acid nucleation. For simplification, only evaporation of one ammonia ($\gamma_{B,AB}$) or one sulfuric acid ($\gamma_{A,AB}$) molecule was taken into consideration. All evaporation rates were set to zero to calculate nucleation at the kinetic limit. Clusters with five and more sulfuric acid
355   molecules were considered to have an equal number of acids and bases.



Wall loss was calculated as

$$k_{wall} = C\sqrt{D} \qquad (16)$$

where $C$ is a chamber specific constant and depends on chamber dimensions and air mixing (fan speed). $C$=0.77 cm$^{-1}$ s$^{-0.5}$ was used in this study based on a measured sulfuric acid wall loss rate of 0.002 s$^{-1}$. $D$ is the diffusion constant calculated as

$$D = \frac{kTC_c}{3\pi\eta d_p} \qquad (17)$$

where $k$, $T$, $C_C$, $\eta$, $d_p$ are the Boltzmann constant, temperature, Cunningham slip correction factor, gas viscosity and particle diameter, respectively. When discussing effects of condensation sink, condensation sinks were considered as being composed of varying concentrations of particles with a diameter of 100 nm.

Author Contribution: M.X., D.S., A.K., M.W., H.L., B.M.,U.M., A.B.,M.S., X.-C.H., K.L.,L.A., R.B., D.C., A.D., J.Du., H.F., V.H., C.K., T.K., J.L. C.L., Z.L., H.M., V.M., H.E.M., R.L.M., A.O., E.P., T.P., J.P., V.P., L.Q., M.R., Sie.S., Sim.S., Y.S.,Y.J.T., A.T., M.V., A.W., R.W., Yo.W., L.W., D.W., Yu. W., C.Y., P.Y., Y.Y., Q.Z., X.Z. R.F., M.K., J.K., J.Do. prepared the CLOUD facility and measurement instruments. X.M., C.R.H., L.D., H.L., U.M., A.B., T.K., M.R., A.T., L.W., A.H., K.C., V.R., R.C.F., M.K., D.W., J.K., N.M.D., U.B., I-E.H., J.Do. planed the experiments. M.X.,C.R.H., L.D., D.S., M.W., H.L.,O.G. B.M.,U.M., A.B.,M.S., X.-C.H., K.L., L.A., R.B., P.S.B., L.B., D.B., F.B., S.B.,R.C., A.D., J.Du., H.F., V.H., C.K., J.L., C.P.L., Z.L., V.M., R.M.,R.L.M.,W.N., E.P., J.P., V.P., M.R., Sim.S., Y.J.T.,A.T., M.V., A.W.,L.W., D.W., Yu.W., C.Y., P.Y.,Q.Y., X.Z.,A.A.,P.M.W., I-E.H., J.Do. collected the data. X.M., C.R.H., L.D., D.S., A.K., M.W., H.L., O.G., B.M., U.M., M.S., H.F., C.K., R.L.M., A.C.W., L.W., P.Y., R.F., J.D. analyzed the data. X.M., C.R.H., D.S., M.W., H.L., O.G., U.M., A.B., F.B., H.G., C.K., T.P., M.R., Sie.S., P.Y., J.C., A.H., R.V., R.C.F., K.M., D.W., J.K., N.M.D., U.B., I-E.H., J.Do. contributed to the scientific discussion. M.X., C.R.H, H.L., T.P., R.F., N.M.D., U.B., I-E.H., J.Do. contributed to the writing of the manuscript.

Data availability: Data related to this article are available upon request to the corresponding authors.
Supplement.

Competing interests: The authors declare that they have no conflict of interest.

**Acknowledgements**
We thank CERN for supporting CLOUD with technical and financial resources, and for providing a particle beam from the CERN Proton Synchrotron. We thank tofTools team for providing programs for mass spectrometry analysis. We thank P. Carrie, L.-P. De Menezes, F. Josa, I. Krasin, O.S. Maksumov, M.V. Philippov, R. Sitals, A. Wasem, K. Ivanova for their contributions to the experiment. Funding: This research has received funding from the EC Seventh Framework Programme and European Union's Horizon 2020 programme (Marie Skłodowska Curie ITNs no. 316662 "CLOUD-TRAIN" and no.



764991 "CLOUD-MOTION"), Horizon 2020 (Marie Skłodowska-Curie Grant "Nano-CAVa" 656994), Horizon 2020 MC-COFUND Grant (665779), ERC Advanced ('ATM-GP' grant no. 227463), ERC-Consolidator Grant (NANODYNAMITE 616075), ERC-Starting grant (COALA, grant no. 638703, QAPPA, grant no. 335478), the Swiss National Science Foundation (no. 200021_169090, 200020_172602, 20FI20_172622), the U.S. National Science Foundation (grants AGC1439551, AGS1447056, AGS1531284, AGS1801574, AGS1801897, AGS1649147, AGS1801280, AGS1602086, 1801329), Wallace Research Foundation, German Federal Ministry of Education and Research(01LK1222A CLOUD-12 and 01LK1601A CLOUD-16), the Portuguese Foundation for Science and Technology (project no. CERN/FIS-COM/0014/2017), the Presidium of the Russian Academy of Sciences ("High energy physics and neutrino astrophysics" 2015), the Presidium of the Russian Academy of Sciences (the Program "Physics of Fundamental Interactions" 2017-2020), Austrian Science Fund (FWF, project number J3951-N36), the Austrian Science Fund (FWF; project no. P27295-N20), NASA graduate fellowship (NASA-NNX16AP36H), Academy of Finland (project number 299574, 307331 and 310682), CERN/FIS-COM/0014/2017.

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





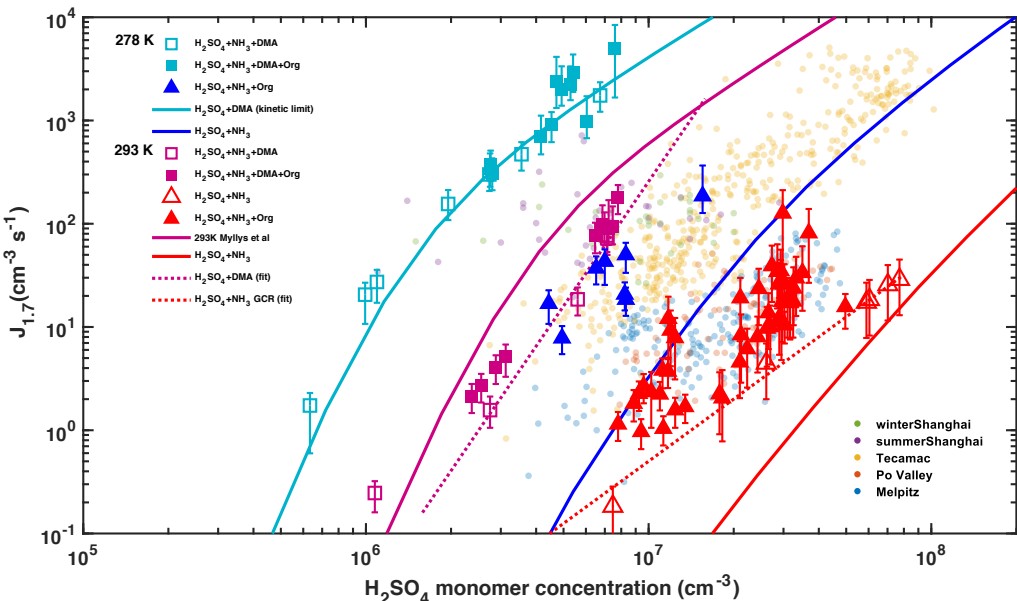

**Figure 1: Dependencies of atmospheric particle formation rates ($J_{1.7}$) on sulfuric acid.** $J_{1.7}$ versus $H_2SO_4$ measured in CLOUD at 278 K (blue and cyan) or 293 K (red and magenta) with 1 to 2 ppbv $NH_3$. Experiments without DMA injection are shown by triangles and those with 4 pptv DMA by squares. Filled and open symbols indicate presence or absence of anthropogenic organics, respectively, where the variability of the filled symbols is mostly due to different concentrations of oxidised organics (OxOrg) from NAPH, TMB and TOL oxidation, and $NO_x$. Over the range of $10^6$ to $10^8$ cm$^{-3}$ sulfuric acid, $J_{1.7}$ increases from 1 to $10^4$ cm$^{-3}$s$^{-1}$. The error bars indicate the measurement uncertainty of the nucleation rates. Atmospheric observations in the polluted boundary layer are indicated by small coloured circles (Kuang et al., 2008; Paasonen et al., 2010; Yao et al., 2018). Solid lines show the predicted nucleation rates of sulfuric acid and 2 ppbv ammonia at 278 K (blue) and 293 K (red) from the kinetic model. The nucleation rate of $H_2SO_4$-$NH_3$ under GCR conditions is given by the red dashed line. The nucleation rate of $H_2SO_4$ at the kinetic limit is indicated by the solid cyan curve. The solid magenta line is the predicted nucleation rates of sulfuric acid and 4 pptv DMA at 293K and the magenta dashed line shows a fit of $J_{1.7} = k\,[H_2SO_4]^4$ to the data with 4 pptv DMA at 293K.



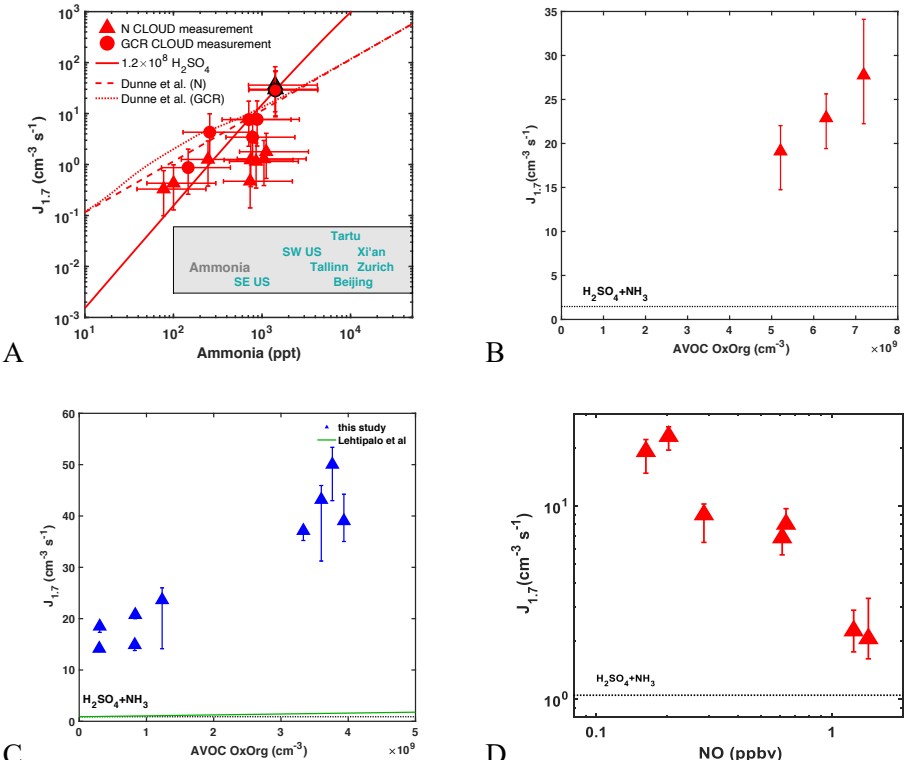

**Figure 2: Dependencies of atmospheric particle formation rates ($J_{1.7}$) on ammonia, organics and NO.** **(A)** $J_{1.7}$ versus $NH_3$ at fixed $H_2SO_4$ ((1.1-1.6) × $10^8$ cm$^{-3}$) and temperature (292-295 K). Red triangles are previous CLOUD measurements (Dunne et al., 2016) under neutral (triangle) or GCR (circle) conditions while the symbols with black outlines are new from this study. Error bars describe run-to-run repeatability of $J$ in CLOUD, which is ca. 30%. The solid red line presents $J_{1.7}$ for 1.2 × $10^8$ cm$^{-3}$ $H_2SO_4$ at 293 K from our kinetic model simulation and fits well to the additional data at the high $NH_3$ mixing ratio. The dashed (N, neutral ) or dotted (GCR) red line shows the parameterisation used by Dunne et al. (2016). Also shown are the mean concentrations of ammonia at various locations (Elser et al., 2018; Guo et al., 2017) as text of locations. **(B)** Experiments with 2.8±0.3 × $10^7$ cm$^{-3}$ $H_2SO_4$, 849±20 pptv $NH_3$ and 0.19±0.03 ppbv NO at 293 K with NAPH:TMB:TOL of 1:2:10. Black dotted line in the plot displays $J_{1.7}$ for 1.9×$10^7$ $H_2SO_4$ nucleating with 1000 pptv $NH_3$. **(C)** Experiments with 8±1.5 × $10^6$ cm$^{-3}$ $H_2SO_4$, 1350±350 pptv $NH_3$ and 0.51±0.3 ppbv NO at 278 K, with NAPH:TMB:TOL of 1:5:30. Black dotted line in the plot displays $J_{1.7}$ for 8 × $10^6$ cm$^{-3}$ $H_2SO_4$ nucleating with 1600 pptv $NH_3$. Green solid line shows $J_{1.7}$ of 8×$10^6$ cm$^{-3}$ $H_2SO_4$ nucleating with 1600 pptv $NH_3$, with BVOC OxOrg and with 0.2 ppbv NO based on parameterizations in Lehtipalo et al. (2018). Oxidation of aromatic compounds by OH forms highly oxygenated molecules (HOMs) (Molteni et al., 2018; Wang et al., 2017) through autoxidation, leading to significant NPF, at rates even higher than observed for monoterpene oxidation. This is consistent with the finding that HOMs from aromatic compounds are less volatile than HOMs from monoterpene precursors at the same number of carbon atoms in the backbone (Wang et al., 2020). **(D)** $J_{1.7}$ versus NO at 293K, with fixed conditions for other vapours (2.3±0.4) ×$10^7$ cm$^{-3}$ $H_2SO_4$, 800 to 1000 pptv ammonia, and an AVOC mixture yielding an OxOrg concentration of (5.2±0.7) ×$10^9$ cm$^{-3}$ (among which (0.9±0.2) ×$10^9$ cm$^{-3}$ produced by NAPH oxidation, (2.1±0.3) ×$10^9$ cm$^{-3}$ by TMB oxidation and (2.2±0.3) ×$10^9$ cm$^{-3}$ by TOL oxidation). The suppression of $J_{1.7}$ with increasing NO is due to reduced production of extremely low volatility OxOrg. The error bars describe the variation of the measurements only, and do not include a 30% run-to-run variability. The horizontal dotted line indicates $J_{1.7}$ for 2.3×$10^7$ cm$^{-3}$ $H_2SO_4$ with 1000 pptv $NH_3$ at 293 K under GCR conditions. $J_{1.7}$ decreases with increasing NO since NO suppresses both autoxidation and the formation of low-volatility dimers. This efficient suppression of aromatic $RO_2$ isomerization to form autoxidation HOM products is in contrast to aromatic $RO_2$ isomerization rates to form bicycloalkylradicals, which dominate the early stage of aromatic oxidation and outcompete NO reactions even at much higher NO concentrations (10s to 100s of ppbv) (Volkamer et al., 2002).

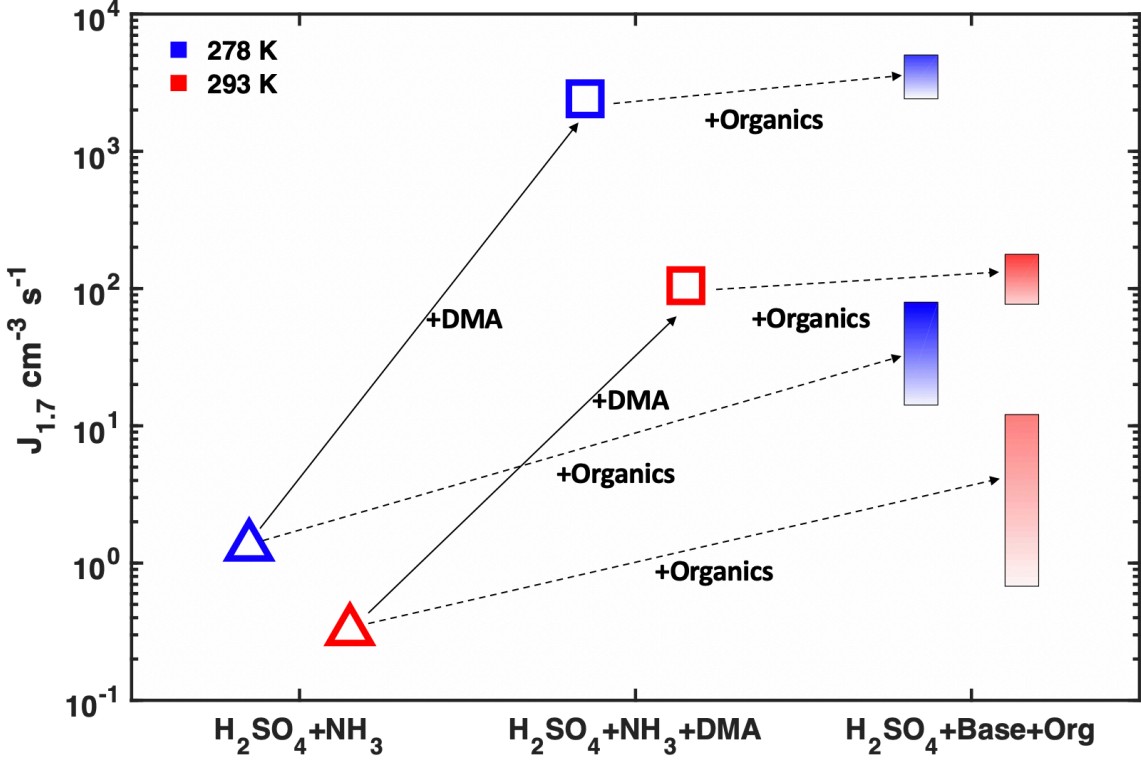


**Figure 3. Figure 3: Simplified scheme of NPF rates in polluted environments showing the effect of adding different vapours on H$_2$SO$_4$-NH$_3$ nucleation at two different temperatures.** All points have similar H$_2$SO$_4$ (8±2 × 10$^7$ cm$^{-3}$) and NH$_3$ (1-2 ppbv). The leftmost points are measured with only SO$_2$ and NH$_3$ added to the chamber, and each step to the right represents the addition of one more component to the system. Solid arrows describe the addition of ca. 4 pptv of DMA, dashed arrows describe the addition of aromatic hydrocarbons. Increasing color intensity of bars indicates increasing production rate of highly oxidised organics.


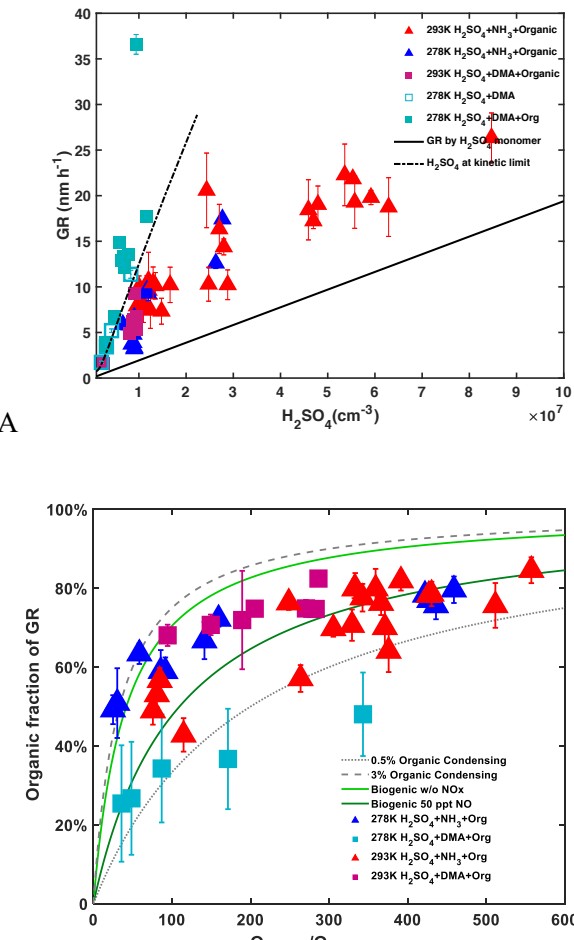

**Figure 4: Influence of sulfuric acid, OxOrg and DMA on growth rate (GR): (A)** GRs are measured between 1.8 nm and 3.2 nm, at 278 K (blue and cyan) or 293 K (red and magenta) with 1 to 2 ppbv $NH_3$ and plotted versus sulfuric acid monomer concentration. GRs without DMA injection are shown by triangles and GRs with 4 pptv DMA by squares. Filled and open symbols indicate presence or absence of organics, respectively, where the variability of the filled symbols is mostly driven by the concentrations of oxidised organics (OxOrg) and

$NO_x$. The solid line shows GR for the condensation of sulfuric acid monomer only while the dashed line displays sulfuric acid nucleating at the kinetic limit, when cluster coagulation dominates the growth. **(B)** Contribution of oxidised organics (OxOrg) to sub-3 nm GR as a function of the ratio of the production rates of oxidised organics ($Q_{OxOrg}$) to sulfuric acid ($Q_{sulfuric\ acid}$). This ratio is roughly proportional to the ratio of condensable organics to sulfuric acid in the CLOUD chamber. The grey lines encompass most data and indicate that 0.5% (dotted) to 3% (dashed) of the total oxidised organics contribute to the growth of sub-3 nm particles. With the exception of sulfuric acid +

DMA at 278 K, organics are the dominant contributor to particle growth in the sub-3 nm size range. At larger particle sizes, the contribution of organics will increase further. Data are compared to the parameterisations of monoterpene oxidation by Lehtipalo et al. (2018) where the dark green line shows a strong decrease in the yield of condensable gases in the presence of only 50 pptv NO compared to the light green line for conditions without NO. Condensable OxOrg yields are generally higher from AVOCs than those obtained from biogenic vapours such as monoterpenes at elevated $NO_X$ conditions (green lines), again consistent with the lower volatility of aromatic OxOrg formed by

multi-generation oxidation in comparison with biogenic (monoterpene) OxOrg with the same oxygen content (Wang et al., 2020a).

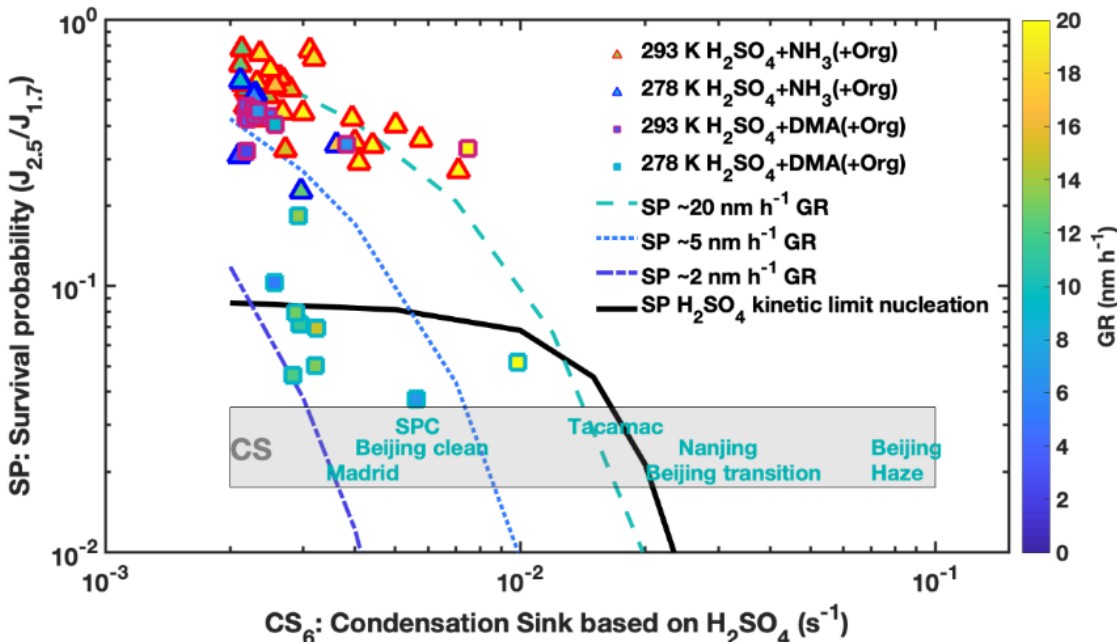

**Figure 5: Influence of OxOrg and DMA on survival probability:** Survival probability (SP) vs condensation sink (CS$_6$). The survival
probability (SP) is calculated as the ratio between the formation rates at 2.5 nm and 1.7 nm. CLOUD data were recorded at 60 % RH and
278 K (blue and cyan contoured) or 293 K (red and magenta contoured) with 1 to 2 ppbv NH$_3$. Experiments without DMA injection are
shown by triangles, and experiments with 4 pptv DMA by squares. The sub-3 nm growth rate is given by the colour of the symbols. The loss
rates of newly formed particles are approximated by the condensation sink of sulfuric acid to particles larger than 6 nm (x-axis). Condensation
sinks observed in the atmosphere are indicated as text of the location (Cai et al., 2017; Carnerero et al., 2018; Kontkanen et al., 2016; Kuang
705  et al., 2010; Yu et al., 2016). Dashed lines are calculated survival probabilities of 2.5 nm particles with growth rates of 2, 5, and 10 nm h$^{-1}$.
The solid line is the survival probability of 2.5 nm particles if 8 x 10$^6$ cm$^{-3}$ sulfuric acid nucleates at the kinetic limit, which would e.g. at
CS$_6$ = 0.002 s$^{-1}$ (representing wall loss) correspond to a nucleation rate $J_{1.7}$ of 4000 cm$^{-3}$ s$^{-1}$ (Fig. 1) and a growth rate of 11 nm h$^{-1}$ (Fig. 4A).
The high production rate of nucleating clusters significantly reduces the survival probability, due to cluster-cluster collisions. However, at
the same time, this increases the growth rate such that the survival probability is unaffected by the CS until it becomes comparable to the
710  loss rate from cluster-cluster collisions.

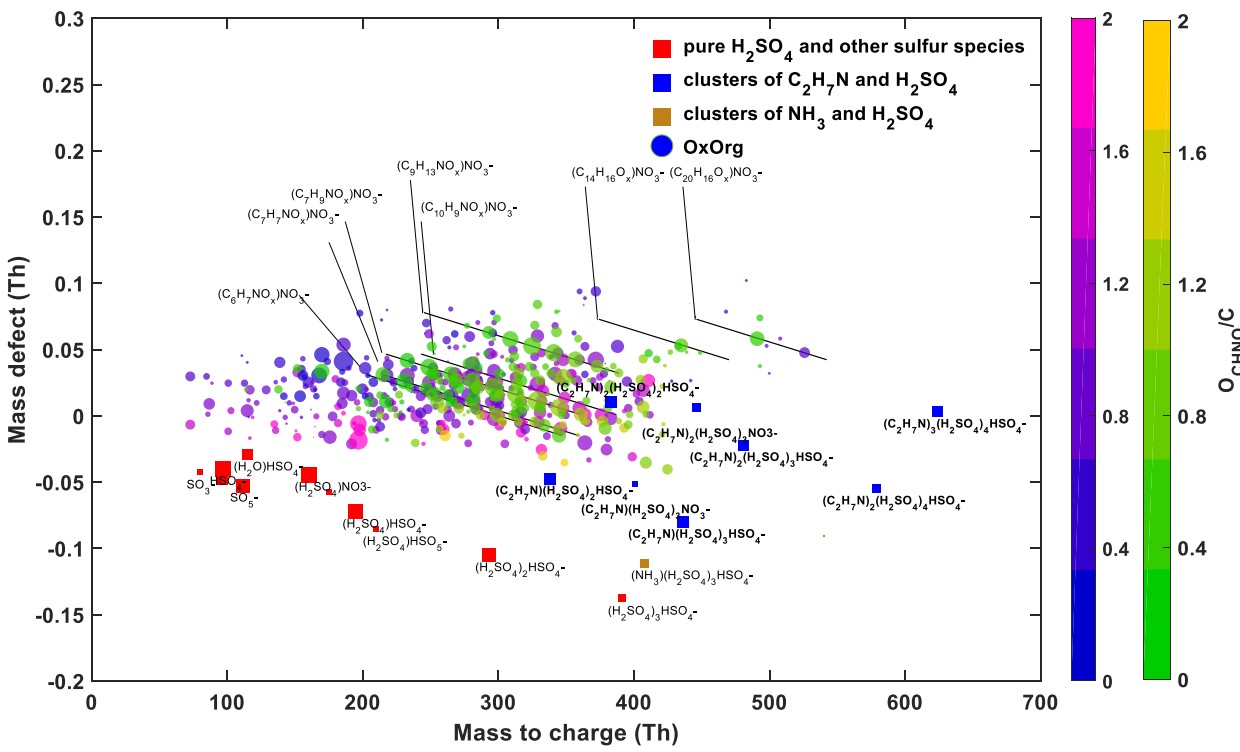

**Figure 6. Mass defect plot of sulfuric acid nucleation with DMA in the presence of organics at 278 K in the CLOUD chamber.** The conditions are 5.2 x 10$^6$ cm$^{-3}$ H$_2$SO$_4$ monomer, 4 pptv DMA, 1200 pptv NH$_3$, 680 pptv NO, 420 pptv, NAPH, 2200 pptv TMB and 9250 pptv TOL. Sulfate clusters are shown by red squares, sulfate-DMA clusters containing up to 5 sulfur molecules by blue squares, sulfate-NH$_3$ clusters by brown squares, and organics by circles coloured by O:C and by their chemical composition CHO or CHNO, respectively.





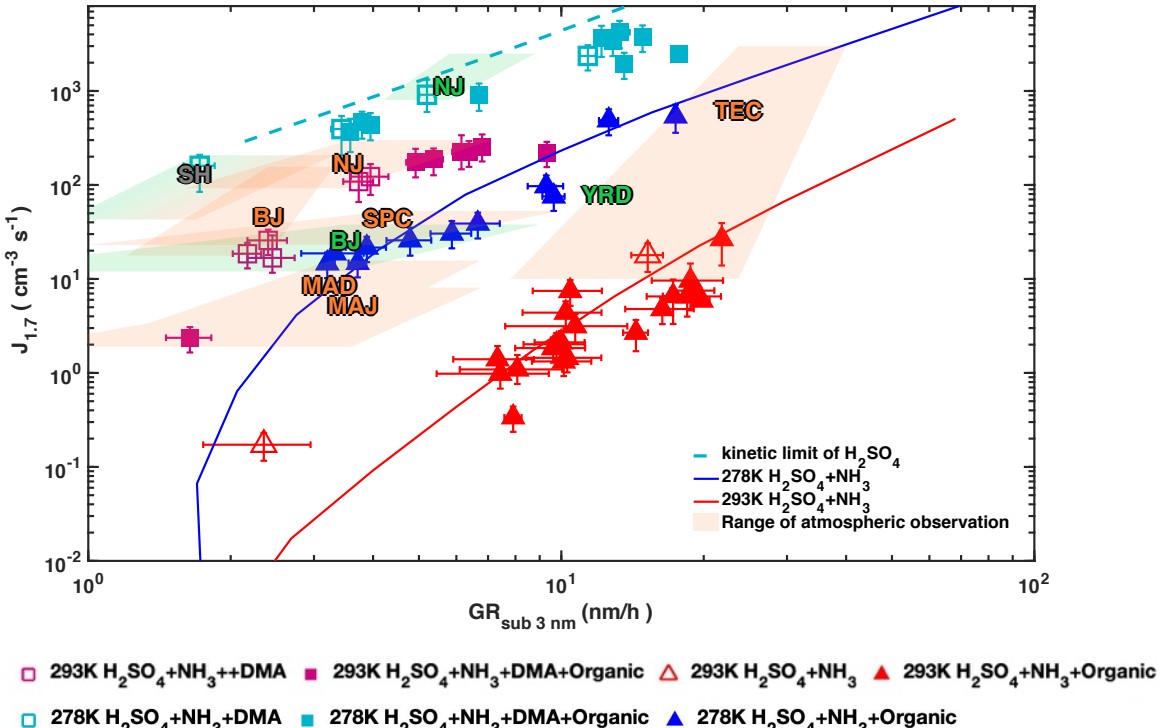

**Figure 7: Particle formation rates ($J_{1.7}$) versus growth rates of sub-3 nm particles. CLOUD data were recorded at 278 K (blue and cyan symbols) or 293 K (red and magenta) with 1 to 2 ppbv NH₃.** Experiments without DMA are shown by triangles and experiments with 4 pptv DMA by squares. Filled and open symbols indicate presence or absence of organics, respectively. Error bars denote uncertainties of $J_{1.7}$ and GR values. Solid lines represent the simulated nucleation/growth rates with a kinetic model of sulfuric acid with 1 ppbv ammonia at 278 K (blue) and 293 K (red). H₂SO₄ nucleation at the kinetic limit is indicated by the cyan dashed line. Also shown are the mean values for sub-3 nm GR and sub-2 nm $J$ from observations in the polluted boundary layer at the following locations: Shanghai (SH) (Yao et al., 2018), Beijing spring and winter(BJ) (Cai et al., 2017; Jayaratne et al., 2017), Madrid (urban: MAD and suburban: MAJ) (Carnerero et al., 2018), Po Valley regional (San Pietro Capofiume, SPC) (Kontkanen et al., 2017), Nanjing summer and winter (NJ) (Yu et al., 2016), Yangtze-River-Delta regional (YRD) (Dai et al., 2017) and Tecamac (TEC) (Kuang et al., 2008). The text colours indicate <288 K (cyan), >288 K (orange) or all year (grey). The shaded area colours indicate the range of ($J$, GR) values at these locations for temperatures <288 K (cyan) and >288 K (orange).