# Peer review of "The driving factors of new particle formation and growth in the polluted boundary layer"

_Atmospheric Chemistry and Physics, 2020_

## Author Comment (AC1)

Response to Review of "The Driving Factors of New Particle Formation and Growth in the Polluted Boundary Layer" by M. Xiao et al.

This is a well-written and extremely interesting manuscript describing a comprehensive set of experiments on the chemical mechanisms of new particle formation (NPF) in an environment representative of a polluted urban atmosphere. The study provides an explanation for the occurrence of NPF when there is very strong competition for precursor vapors by high concentrations of pre-existing aerosol particles. The successful nucleation and growth of particles in these environments is important, because it distributes aerosol mass to smaller sizes, changing the interaction of these particles with sunlight, the numbers of cloud-nucleating particles, and the fate of the aerosol mass in the human respiratory system. These topics have broad interest, and the manuscript is definitely suitable for publication in ACP.

The investigators demonstrate that sulfuric acid + dimethylamine and other bases like ammonia dominate NPF in polluted atmospheres, even in the presence of high concentrations of oxidation products of aromatics, which can have quite low volatility. The growth of the newly formed particles to larger sizes involves many other compounds, including organics. There is a very strong temperature dependence to the NPF.

This work employs cutting-edge research instruments and the CERN CLOUD chamber to make these measurements. The results are well described and the interpretation is clear. The methods are also well described, and earlier work is appropriately referenced and discussed. The manuscript is very well written. I believe it is suitable for publication with only minor technical changes.

The authors would like to thank the reviewer for the detailed comments, corrections, and suggestions. Here we replied to all the comments and improved the paper following their recommendations.

Comments:

1) Throughout the PDF manuscript, following a capital "A" there is a space and the remainder of the word is shifted to the right. This must be some sort of PDF conversion problem, but the authors and copy editors should carefully proof the online and print versions of the final manuscript to see if this issue still exists.

We did not find this feature in the pdf version downloaded from ACP. In any case we will carefully read the proof and focus on this issue.

2) It's a bit odd in ACP to have the Methods following the Results. This is more common in journals such as Nature and Science, where a wide audience is presumed and the methods are relegated to an appendix-like attachment, with a smaller font. I don't have a problem with this structure for ACP, but wish to call attention to it in the event there are editorial norms that would suggest moving the Methods toward the front.

We submitted the manuscript for consideration as an ACP Letter. This requires the description of the applied methods in the form of an appendix. If the manuscript will be accepted as ACP Letter this formatting does comply with the formatting requirements. Otherwise, it will be changed.

3) Line 134: define "GR"

Added: growth rate (GR)

4) Line 138: For clarity, perhaps say "cluster self-coagulation dominates the growth." It took me a moment to understand what the authors were meaning.

We changed the text as the reviewer suggested.

5) Line 144: "The multicomponent system used in the simulations. . . ." What simulations? Stolzenburg et al. (2020)?

It refers to the multicomponent experiments in this study. We changed "the simulations" to "this study".

6) Line 168: The chamber simulations do reproduce observed urban GRs; however, it is probably worth noting here that the more complex chemistry of the ambient urban atmosphere may lead to other species contributing to particle growth than are investigated at the CLOUD chamber. One should not presume that the relatively simple mixtures explored here can replicate the full complexity of atmospheric processes.

We agree that the chamber does not represent the full complexity in the atmosphere. However, the multicomponent system we used reproduce the major variabilities in terms of particle production rates, particle growth rates, sulfuric acid and bases concentrations and particle condensation sink. In this regard, in line 164 we state: 'While the precursor composition in the ambient atmosphere is indeed much more complex than in the chamber'. Details on the selection of organics and oxidation processes are further discussed at line 250-260 and SI. We think this covers the precaution raised by the reviewer.

7) Line 214: In the previous paragraph you state that the experiments were at the CLOUD chamber; no need to repeat that here.

The repetition is removed.

8) Line 240: DMA is already defined.

We removed the repeated definition.

9) Line 244: No mention here of how OH was determined; this appears later in the manuscript (Eq. 5)but would logically be verbally described here.

We think the detailed description of OH and OxOrg determination would break the flow of text. We add (for details see section 4.5).

10) Line 265: Replace "DMA-train" with "differential mobility analyzer train". Too many "DMA"s.

DMA is replaced with the full name "differential mobility analyzer", also later in the text

11) Lines 266-7: Define PSM, SMPS, and CPC.

PSM and CPC are already defined earlier in this paragraph. Scanning mobility particle sizer is added for SMPS.

12) Eq. 1: Perhaps have a subscript "dp" for the J and N variables, since you calculate these values for different size particles (e.g., 1.7 nm, 3 nm).

"$J$" changed to "$J_{dp}$" and "$N$" changed to '$N_{dp}$".

13) Line 305: Provide model number and company name for the H3O+ CIMS and state the method (e.g., cavity ringdown spectrometry), model number, and company name for the NH3 analyser.

The $H_3O^+$ CIMS is an APi-TOF (TOFWERK AG) coupled with a home-made crossflow ionization source. This is further specified as follows:

"Ammonia concentrations were measured with cavity ring-down spectroscopy (G2103, Picarro, Inc) and an $H_3O^+$ chemical ionisation mass spectrometer ($H_3O^+$ CIMS) (Pfeifer et al., 2020). The latter was an APi-TOF (TOFWERK AG) coupled with a home-made crossflow ionization source using positively charged water clusters to detect ammonia in real time."

14) Line 311: Is the PTR3 a custom-built instrument? If so, say this and if not, give model number and company name.

PTR3 is custom-made. We add "custom-made" in the description.

15) Line 314: Define STOF and give model number and company name for the "PTRS", which should be "PTRMS".

PTRS is also custom-made. We changed the text as "… a custom-made short TOF proton transfer mass spectrometer was used in CLOUD12".

16) Line 359: Please reference the form of the Cunningham slip correction equation you are using; there are several and they do have some differences.

We add the Cunningham slip correction equation we used as equation 18.

$$C_c = 1 + Kn * (1.142 + 0.558 * exp(-0.999/Kn));$$

17) References: Please review the reference formatting and ensure it follows Copernicus guidelines. EndNote-style reference managers always make mistakes; for example, for Breitenlechner et al. the title of the paper is capitalized, and in Dunne et al. the page range is not completed. If you correct these errors now it will save the copy editors the effort of finding all them and asking you to fix them later.

We thank the reviewer for spotting these errors. We have corrected these references.

18) Fig. 1. I used a pen to label all the lines and symbols because this is such a busy graph. I suggest you go ahead and do that to make it much easier to interpret.

We added the labels to guide the eyes.

[Figure]

19) Fig. 1 caption. The penultimate sentence should say that the nucleation rate of H2SO4+DMA at the kinetic limit is shown by the cyan curve.

The cyan curve shows the $H_2SO_4$ nucleation rate at the kinetic limit, which matches our $H_2SO_4$+DMA experiments. The line is not specifically modeled for $H_2SO_4$+DMA and the kinetic limit is not sensitive to the presence or absence of DMA. We changed the sentence to "The nucleation rate of $H_2SO_4$ at the kinetic limit is indicated by the solid cyan curve, which matches our $H_2SO_4$+DMA experiments."

20) Fig. 2 caption: In the last sentence there needs to be a space in bicycloalkylradicals.

We added the space, now it is "bicycloalkyl radicals".

21) Fig. 4 caption: What do you mean by, "At larger particle sizes, the contribution of organics will increase further."? What's the basis for this statement.

As particle size increases, more volatile organics can also participate in particle growth. We added Tröstl et al., 2016 here.

22) Fig. 4b: It's hard to distinguish the green and blue curves (biogenic with and without NO) from each other. Can you use a different line type for each?

We changed biogenic without NO to a dashed line.

23) Fig. 5: In the caption please list the name and country of each location indicated in the condensation sink box in the graph, as you did for Fig. 7.

We added the following text to the caption: "Also shown are typical CS from observations in the polluted boundary layer at the following locations: Po Valley regional (San Pietro Capofiume,

SPC) (Kontkanen et al., 2017), Madrid (Carnerero et al., 2018), Tecamac (Kuang et al., 2010), Nanjing (Yu et al., 2016) and Beijing (clean, transition and haze) (Cai et al., 2017)."

24) Fig. 6. What information is conveyed by the size of the symbols?

 The size of the symbols is proportional to their intensity in the mass spectrum. We added this information in the caption.

Reference:

Cai, R., Yang, D., Fu, Y., Wang, X., Li, X., Ma, Y., Hao, J., Zheng, J. and Jiang, J.: Aerosol surface area concentration: a governing factor in new particle formation in Beijing, Atmos. Chem. Phys., 17(20), 12327–12340, doi:10.5194/acp-17-12327-2017, 2017.

Carnerero, C., Pérez, N., Reche, C., Ealo, M., Titos, G., Lee, H. K., Eun, H. R., Park, Y. H., Dada, L., Paasonen, P., Kerminen, V. M., Mantilla, E., Escudero, M., Gómez-Moreno, F. J., Alonso-Blanco, E., Coz, E., Saiz-Lopez, A., Temime-Roussel, B., Marchand, N., Beddows, D. C. S., Harrison, R. M., Petäjä, T., Kulmala, M., Ahn, K. H., Alastuey, A. and Querol, X.: Vertical and horizontal distribution of regional new particle formation events in Madrid, Atmos. Chem. Phys., 18(22), 16601–16618, doi:10.5194/acp-18-16601-2018, 2018.

Kontkanen, J., Lehtipalo, K., Ahonen, L., Kangasluoma, J., Manninen, H. E., Hakala, J., Rose, C., Sellegri, K., Xiao, S., Wang, L., Qi, X., Nie, W., Ding, A., Yu, H., Lee, S., Kerminen, V.-M., Petäjä, T. and Kulmala, M.: Measurements of sub-3 nm particles using a particle size magnifier in different environments: from clean mountain top to polluted megacities, Atmos. Chem. Phys., 17, 2163–2187, doi:10.5194/acp-17-2163-2017, 2017.

Kuang, C., Riipinen, I., Sihto, S.-L., Kulmala, M., Mccormick, A. V and Mcmurry, P. H.: An improved criterion for new particle formation in diverse atmospheric environments, Atmos. Chem. Phys., 10, 8469–8480, doi:10.5194/acp-10-8469-2010, 2010.

Pfeifer, J., Simon, M., Heinritzi, M., Piel, F., Weitz, L., Wang, D., Granzin, M., Müller, T., Bräkling, S., Kirkby, J., Curtius, J. and Kürten, A.: Measurement of ammonia, amines and iodine compounds using protonated water cluster chemical ionization mass spectrometry, Atmos. Meas. Tech., 13(5), 2501–2522, doi:10.5194/amt-13-2501-2020, 2020.

Tröstl, J., Chuang, W. K., Gordon, H., Heinritzi, M., Yan, C., Molteni, U., Ahlm, L., Frege, C., Bianchi, F., Wagner, R., Simon, M., Lehtipalo, K., Williamson, C., Craven, J. S., Duplissy, J., Adamov, A., Almeida, J., Flagan, R. C., Franchin, A., Fuchs, C., Guida, R., Gysel, M., Pet, T. and Steiner, G.: The role of low-volatility organic compounds for initial particle growth in the atmosphere, Nature, 533, 527–531, doi:doi:10.1038/nature18271, 2016.

Yu, H., Zhou, L., Dai, L., Shen, W., Dai, W., Zheng, J., Ma, Y. and Chen, M.: Nucleation and growth of sub-3 nm particles in the polluted urban atmosphere of a megacity in China, Atmos. Chem. Phys., 16, 2641–2657, doi:10.5194/acp-16-2641-2016, 2016.

---

## Author Comment (AC2)

Response to the comments of referee II on the driving factors of new particle formation and growth in the polluted boundary layer by Xiao et al.

Xiao et al. make an important contribution on nailing down the different role of precursors and environment factors (mainly temperature) on new particle formation and growth in contrast/diverse conditions in the atmosphere. In their controlled chamber experiments, large variability in NPF rate has been observed, which covers the observed variability in the real atmosphere. The results and conclusion will be very helpful on understanding and interpretating the ambient observations of NPF rates and growth, especially in polluted urban environment. The manuscript is well written, and plots are nicely crafted.

I recommend its publication and have following comments for improvements.

The authors would like to thank the reviewer for the constructive comments, corrections, and suggestions. Here we replied to all the comments and improved the paper following his/her recommendations.

As the main goal of this study is to explain the large variability of the NPF rate in the different environment, it would be nice if the related real atmospheric environment that can be clearly defined, for example, Beijing clean, Madrid, Nanjing, Nanjing Beijing transition, Beijing haze, SE US, SW US, Tartu, Xian, Zurich and so on. I see that the authors have somehow made such attempt by marking them in some plots (e.g., Figure 2A and Figure 5 etc.), but a table of concentrations ranges of the varied precursors, as well as the ambient temperature when the observations were made would be very helpful. I also suggest strengthening a bit the discussion on comparison between chamber conditions and real environmental conditions.

We added a table containing nucleation rates, growth rates, condensation sinks, temperature, and precursors as the reviewer suggests to the SI. Atmospheric parameters span a wide range and we cover this parameter space within the chamber capability to a great extent.

| | J (cm$^{-3}$ s$^{-1}$) | GR (nm h$^{-1}$) | H$_2$SO$_4$ (cm$^{-3}$) | NH$_3$ | Amine | Org (cm$^{-3}$) | season | Temp (℃) | CS (s$^{-1}$) |
|---|---|---|---|---|---|---|---|---|---|
| Tecamac[a] | 162 (17~20 57) | 20 (7.8~> 39) | 7×10$^6$-1.7×10$^7$- 6.6×10$^7$ | | | | end of March to early April | 10-29 | 0.01-0.035 |
| Beijing, Spring[b] | 46.4 (17.7- 156) | 2.4 (1.2- 3.3 ) | 2×10$^5$-1.6×10$^6$ | 10 ppb[d] | | | March | 8-24 | 0.0028 (clean), 0.02 (transition), 0.07 (haze) |
| Beijing, Winter[c] | 26(12- 38) | 3.5 (0.5-9) | | | | | end of October to January | -15- 20 | 0.0042 (0.0023- 0.0057) |
| Beijing[e] | 3-100 | | 3×10$^6$-1×10$^7$ | | 5-32 ppt | | All year round | | 0.017 |
| Nanjing, warm[f] | 92-300 | 1.6-7.1 | 1.3×10$^7$-3.1×10$^7$ | | | 1.7×10$^7$- 8.5×10$^7$ | | | 0.016-0.019 |
| Nanjing, cold[f] | 190- 2500 | 4.2-8.8 | 3.5×10$^7$-4.8×10$^7$ | | | 1.1×10$^8$- 1.7×10$^8$ | | | 0.028-0.033 |
| YRD[g] | 80 | 12.5 | | | | | winter | | 0.026±0.014 |

| | | | | | | | | |
|---|---|---|---|---|---|---|---|---|
| SPC[h] | 45 (23-53) | 4.3 (1-10) | | | | | 20-35 | 0.005-0.025 |
| Madrid[i] | 1.1-6.7 | 2.4-15.7 | | | | | summer | 3.40E-03 |
| MAJ[i] | 0.7-6.8 | 1.9-8.1 | | | | | | 2.50E-03 |
| Shanghai[j] | 105 (42-207) | 1.4 (0.55-3.3) | $4\times10^6$-$2\times10^7$ | | ~100 pptv | | All year round | ~0.02 |
| CLOUD $H_2SO_4$+DMA 278 K[h] | 160-2475 | 1.72-18 | $6.3\times10^5$-$7.6\times10^6$ | 1-2.5 ppb | 4 pptv | HOMs up to $5\times10^7$ OxOrg up to $8.8\times10^9$ | 5 | 0.002-0.01 |
| CLOUD $H_2SO_4$+DMA 293 K[h] | 2.4-231 | 1.6-6.8 | $1\times10^6$-$7.8\times10^6$ | 1-2.5 ppb | 4 pptv | HOMs up to $8\times10^7$ OxOrg up to $6\times10^9$ | 20 | 0.002-0.008 |
| CLOUD $H_2SO_4$+$NH_3$ 278 K[h] | 15-537 | 3.2-17.4 | $5\times10^6$-$1.5\times10^7$ | 1-2.5 ppb | Not added | HOMs up to $3\times10^7$ OxOrg up to $9.6\times10^9$ | 5 | 0.002-0.004 |
| CLOUD $H_2SO_4$+$NH_3$ 278 K[h] | 0.17-27 | 2.4-22 | $7.4\times10^6$-$7.9\times10^7$ | 1-2.5 ppb | Not added | HOMs up to $3\times10^8$ OxOrg up to $9.2\times10^9$ | 20 | 0.002-0.007 |

[a](Kuang et al., 2008), [b](Cai et al., 2017), [c](Jayaratne et al., 2017), [d](Guo et al., 2017), [e](Cai et al., 2021), [f](Yu et al., 2016), [g](Dai et al., 2017), [h](Kontkanen et al., 2016), [h](Carnerero et al., 2018), [j](Yao et al., 2018), [h]this study.

We have added the following to section 2.3:

"This controlled laboratory exploration spans the relevant ambient conditions, and provides a detailed understanding and constraints of the governing chemical and physical processes of NPF and early growth. This provides the possibility to rationalize the available ambient measurements and a framework to plan future measurements in the best way."

"These conflicting observations are subject of intense scientific debate of late (Brean et al., 2020; Cai et al., 2020; Guo et al., 2020) and highlight…" It would be nice if the authors could elaborate a bit here what are the "scientific debate" here, and I believe it will help the readers to have a better overview and understanding of the motivation of this study.

We added a brief description on the major views of the referenced papers. The modified text reads now:

The high NPF rates, believed to drive haze events in China (Guo et al., 2014), have been associated with the nucleation of sulfuric acid ($H_2SO_4$) in the presence of amines (Yao et al., 2018). In contrast, at other urban locations (Kuang et al., 2008), reported NPF rates are several orders of magnitude lower at similar $H_2SO_4$ concentrations, despite high levels of condensable species able to grow newly formed particles. Cai et al. (2020) attribute NPF in Beijing to $H_2SO_4$-amine cluster formation, which is modulated by coagulation scavenging. In Barcelona, Brean et al. (2020) linked nucleation to sulfuric acid clustering with both highly oxygenated organic molecules (HOMs) and bases, while Guo et al. (2020) argued photooxidation of organics from vehicular exhaust is responsible for the formation of ultrafine particles in Beijing. These conflicting observations and interpretations highlight the need to better understand the role of the different vapours and environmental parameters and

to quantify their relative contribution in new-particle formation and growth in different polluted locations.

I understand that putting the Method section at the end is probably to obey the formatting requirement of letter, the authors may want to refer to sub-sections, descriptions and equations of the Method section in the main text, so the readers could cross check. For example, how are the "Organic fraction of GR" in Figure 4B derived?

We would like to thank the reviewer for this suggestion. We went through the main text and added references to the method section when necessary. In Figure 4 caption, we add "(determined by subtracting GR by sulfuric acid)" to "Contribution of oxidised organics (OxOrg) to sub-3 nm GR".

In Figure 3, organic seems having stronger effects when DMA is absent. Could the authors comment on it?

For clarity, I suggest explaining that "Base" means both NH3 and DMA are present in the figure caption. There is an additional "Figure 3." in the figure caption.

This observation is correct and is mentioned at line 123, where the effect of organics is discussed: "In the presence of amines, the contribution of organics to NPF is marginal, since the inorganic nucleation rate is overwhelming." Since amines are so efficient in stabilizing sulfuric acid, the organic effect becomes marginal. At line 129 it already says: Further addition of organics when DMA is already present will only marginally affect nucleation rates.

"Base" is replaced by "$NH_3$+DMA". We have removed the extra Figure 3.

In Figure 5, survival probability in Beijing haze condition seems extremely low. Here, CS6 is used, and I am wondering how pollute are the Nanjin Beijing transition and Beijing haze condition? E.g., PM1 or PM2.5 concentrations? A related question is that "polluted boundary layer" is with what PM1 or PM2.5 levels approximately?

In Beijing the occurrence of new particle formation is increasingly occurring at a condensation sink lower than 0.03 $s^{-1}$ (Deng et al., 2021). On NPF days a median CS of 0.009 $s^{-1}$ was measured corresponding to a PM2.5 concentration of 10 $\mu g\ m^{-3}$. In Nanjing CS is also lower than 0.03 $s^{-1}$ albeit PM2.5 concentrations are higher on NPF days (in spring up to 112 $\pm$ 68 $\mu g\ m^{-3}$), which could be due to a shift in the particle size distribution to a larger size (Yu et al., 2016). Particle growth to larger sizes is also often not observed.

"Polluted boundary layer" means not only high aerosol concentrations but also other pollutants such as ammonia, amines or volatile vapors. Also "polluted" is a relative term. For example, rural or remote areas could be relatively low in aerosol and anthropogenic organic vapours compared to industrialized/ urban regions, but still polluted due to transport of pollution or agricultural emissions compared to pristine. Thus, PM concentrations in "polluted boundary layer" cover a wide range between a few to tens or hundreds of $\mu g\ m^{-3}$.

We added this sentence at the end of the introduction to outline what we mean by polluted boundary layer: The experiments cover low and moderately polluted rural environments as well as highly polluted urban situations.

"We compare chamber simulations and atmospheric observations in Fig. 7 using sub-3 nm GR as a proxy for total condensable vapour concentrations…" It seems that total condensable vapor

concentration in BJ is not high as marked in Figure 7. One would expect that high as the pollution level there is quite high. Could the authors comment on it?

The concentrations of condensable vapours are controlled by their production and sink. While precursor concentrations are high, radiation in Beijing is not as strong as in the southern cities such as Shanghai or Nanjing. High NOx levels also limit the production of extremely low volatility compounds from organic vapours. Sinks of condensable vapours are also high in Beijing since a high pollution level results in a high condensation sink. As a result, the total concentration of condensable vapours is not extremely high in Beijing.

Reference:

Cai, R., Yang, D., Fu, Y., Wang, X., Li, X., Ma, Y., Hao, J., Zheng, J. and Jiang, J.: Aerosol surface area concentration: a governing factor in new particle formation in Beijing, Atmos. Chem. Phys., 17(20), 12327–12340, doi:10.5194/acp-17-12327-2017, 2017.

Cai, R., Yan, C., Yang, D., Yin, R., Lu, Y., Deng, C., Fu, Y., Ruan, J., Li, X., Kontkanen, J., Zhang, Q., Kangasluoma, J., Ma, Y., Hao, J., Worsnop, D. R., Bianchi, F., Paasonen, P., Kerminen, V., Liu, Y., Wang, L., Zheng, J., Kulmala, M. and Jiang, J.: Sulfuric acid–amine nucleation in urban Beijing, Atmos. Chem. Phys., 21(4), 2457–2468, doi:10.5194/acp-21-2457-2021, 2021.

Carnerero, C., Pérez, N., Reche, C., Ealo, M., Titos, G., Lee, H. K., Eun, H. R., Park, Y. H., Dada, L., Paasonen, P., Kerminen, V. M., Mantilla, E., Escudero, M., Gómez-Moreno, F. J., Alonso-Blanco, E., Coz, E., Saiz-Lopez, A., Temime-Roussel, B., Marchand, N., Beddows, D. C. S., Harrison, R. M., Petäjä, T., Kulmala, M., Ahn, K. H., Alastuey, A. and Querol, X.: Vertical and horizontal distribution of regional new particle formation events in Madrid, Atmos. Chem. Phys., 18(22), 16601–16618, doi:10.5194/acp-18-16601-2018, 2018.

Dai, L., Wang, H., Zhou, L., An, J., Tang, L., Lu, C., Yan, W., Liu, R., Kong, S., Chen, M., Lee, S. and Yu, H.: Regional and local new particle formation events observed in the Yangtze River Delta region, China, J. Geophys. Res., 122(4), 2389–2402, doi:10.1002/2016JD026030, 2017.

Deng, C., Cai, R., Yan, C., Zheng, J. and Jiang, J.: Formation and growth of sub-3 nm particles in megacities: impact of background aerosols, Faraday Discuss., 226, 348–363, doi:10.1039/D0FD00083C, 2021.

Guo, H., Weber, R. J. and Nenes, A.: High levels of ammonia do not raise fine particle pH sufficiently to yield nitrogen oxide-dominated sulfate production, Sci. Rep., 7(1), 12109, doi:10.1038/s41598-017-11704-0, 2017.

Jayaratne, R., Pushpawela, B., He, C., Li, H., Gao, J., Chai, F. and Morawska, L.: Observations of particles at their formation sizes in Beijing, China, Atmos. Chem. Phys., 17, 8825–8835, doi:10.5194/acp-17-8825-2017, 2017.

Kontkanen, J., Järvinen, E., E. Manninen, H., Lehtipalo, K., Kangasluoma, J., Decesari, S., Paolo Gobbi, G., Laaksonen, A., Petäjä, T. and Kulmala, M.: High concentrations of sub-3nm clusters and frequent new particle formation observed in the Po Valley, Italy, during the PEGASOS 2012 campaign, Atmos. Chem. Phys., 16(4), 1919–1935, doi:10.5194/acp-16-1919-2016, 2016.

Kuang, C., McMurry, P. H., McCormick, A. V. and Eisele, F. L.: Dependence of nucleation rates on sulfuric acid vapor concentration in diverse atmospheric locations, J. Geophys. Res., 113(D10), D10209, doi:10.1029/2007JD009253, 2008.

Yao, L., Garmash, O., Bianchi, F., Zheng, J., Yan, C., Kontkanen, J., Junninen, H., Mazon, S. B., Ehn, M., Paasonen, P., Sipilä, M., Wang, M., Wang, X., Xiao, S., Chen, H., Lu, Y., Zhang, B., Wang, D., Fu, Q., Geng, F., Li, L., Wang, H., Qiao, L., Yang, X., Chen, J., Kerminen, V. M., Petäjä, T., Worsnop, D. R., Kulmala, M. and Wang, L.: Atmospheric new particle formation from sulfuric acid and amines in a Chinese megacity, Science (80-. )., 361(6399), 278–281, doi:10.1126/science.aao4839, 2018.

Yu, H., Zhou, L., Dai, L., Shen, W., Dai, W., Zheng, J., Ma, Y. and Chen, M.: Nucleation and growth of sub-3 nm particles in the polluted urban atmosphere of a megacity in China, Atmos. Chem. Phys., 16, 2641–2657, doi:10.5194/acp-16-2641-2016, 2016.